# Comprehensive Evaluation of Genome Gap-Filling Tools Utilizing Long Reads

**DOI:** 10.3390/genes15010127

**Published:** 2024-01-20

**Authors:** Xianjia Zhao, Fang Liu, Weihua Pan

**Affiliations:** 1Zhengzhou Research Base, State Key Laboratory of Cotton Biology, School of Agricultural Sciences, Zhengzhou University, Zhengzhou 450001, China; xianjia@gs.zzu.edu.cn; 2Shenzhen Branch, Guangdong Laboratory of Lingnan Modern Agriculture, Genome Analysis Laboratory of the Ministry of Agriculture and Rural Affairs, Agricultural Genomics Institute at Shenzhen, Chinese Academy of Agricultural Sciences (ICR, CAAS), Shenzhen 518120, China; 3National Key Laboratory of Cotton Bio-Breeding and Integrated Utilization, Institute of Cotton Research, Chinese Academy of Agricultural Sciences (ICR, CAAS), Anyang 455000, China

**Keywords:** gap-filling, long reads, genome assembly

## Abstract

The availability of the complete genome of an organism plays a crucial role in the comprehensive analysis of the entire biological entity. Despite the rapid advancements in sequencing technologies, the inherent complexities of genomes inevitably lead to gaps during genome assembly. To obviate this, numerous genome gap-filling tools utilizing long reads have emerged. However, a comprehensive evaluation of these tools is currently lacking. In this study, we evaluated seven software under various ploidy levels and different data generation methods, and assessing them using QUAST and two additional criteria such as accuracy and completeness. Our findings revealed that the performance of the different tools varied across diverse ploidy levels. Based on accuracy and completeness, FGAP emerged as the top-performing tool, excelling in both haploid and tetraploid scenarios. This evaluation of commonly used genome gap-filling tools aims to provide users with valuable insights for tool selection, assisting them in choosing the most suitable genome gap-filling tool for their specific needs.

## 1. Introduction

The quality of genome assembly significantly impacts the quality of downstream genomic analyses. With the advancement of sequencing technologies, particularly, the emergence of single-molecule sequencing (SMS) technologies, genomes have gradually evolved towards being assembled from telomere to telomere. This has led to an increasing number of complete genomes without gaps, such as those of maize [1], rice [2,3], soybean [4], carrot [5], and others. However, challenges such as uneven sequencing depth, the complexity of genome repeat regions, sequencing errors, and other factors frequently lead to the occurrence of numerous gaps during genome assembly. To fill these gaps, extensive efforts have been made. Some approaches involve using next-generation sequencing data to close those gaps. However, a drawback of this strategy is the short length of reads, which makes assembly challenging and computationally intensive, increasing the likelihood of assembly errors. The development of the SMS technology has now made it the mainstream method for filling gaps in genome assembly. SMS reads are longer, which makes it easier to eliminate gaps with lower computational challenges. Therefore, using SMS reads to fill genome assembly gaps has become the preferred approach for researchers when addressing missing regions in genomes.

The method of filling gaps using SMS reads is primarily based on alignment. Various software tools, such as FGAP [6], LR_Gapcloser [7], TGS-GapCloser [8], PGcloser [9], DENTIST [10], RFfiller [11], and SAMBA [12], utilize these strategies for gap closure. FGAP employs long reads or contigs to fill gaps. It initiates by aligning these sequences to the draft genome through BLAST [13]. The most suitable sequences are then selected for gap closure. LR_Gapcloser can work with both corrected and uncorrected long reads. It segments long reads into uniform-length fragments and aligns them to scaffolds using BWA [14]. Based on the alignment results, it identifies long reads capable of bridging the gaps. TGS-GapCloser is versatile and can handle various types of long reads and contigs generated by assembly software. It begins by identifying gap regions within scaffolds, splitting the scaffolds, and aligning long reads to the divided scaffolds. Candidate sequences are chosen from these alignments and undergo further refinement to select the highest-quality sequence for gap closure. PGcloser also supports both long reads and contigs for gap closure. It identifies anchor points at both ends of the gaps and aligns them to long reads. This process aids in selecting suitable long reads to either fill gaps or reduce the gap size. DENTIST utilizes long reads to close gaps. It starts by identifying and masking repetitive regions. Subsequently, it aligns long reads to scaffolds and derives a consensus from all the reads that can be used to fill the gaps. RFfiller uses both long reads and contigs to close gaps. It begins by creating a Markov chain based on alignment information between contigs, scaffolds, and paired-end reads. This Markov chain is then used to allocate long reads or contigs to the gap regions, effectively closing the gaps. These tools rely on alignment and sequence selection methods to bridge gaps in genomic assemblies, harnessing the potential of single-molecule sequencing data. SAMBA is employed to fill gaps generated during the assembly process using long reads, thereby improving the continuity of the assembly results. It utilizes long reads to reassemble contigs from the existing genome assembly, concurrently filling in the gaps during the reconstruction. When filling gaps in scaffolds, the process involves initially fragmenting the scaffold into contigs, subsequently reconstructing these contigs using long reads, and ultimately re-linking the contigs back into the scaffold. This approach may potentially introduce errors in contigs due to the reconstruction process.

Today, the use of long reads to fill gaps has gradually become the primary choice in genome assembly. Users generally choose gap-filling software based on the type of data they have. However, the selection of the most suitable software to improve the completeness and accuracy of the genome after gap filling requires careful consideration. With numerous software options available for gap filling utilizing long reads, testing each one can be resource-intensive, in terms of both time and computational resources. To address this issue, we conducted tests on several software options. We made efforts to compile a comprehensive selection of gap-filling software, considering factors such as maintenance status and user-friendliness. Ultimately, we selected seven software tools for our study: FGAP, LR_Gapcloser, TGS-GapCloser, PGcloser, DENTIST, RFfiller, and SAMBA. Our investigation involved evaluating the performance of these seven software tools across different ploidy levels. We utilized QUAST [15] for assessment and introduced two new metrics: completeness and accuracy. Additionally, we documented the execution time and memory usage.

## 2. Materials and Methods

### 2.1. Datasets

Our dataset comprised three main components: sequencing or synthetically generated reads, preliminary genome assembly results, and a reference genome. The sequencing reads were sourced from the work of Hou et al. [16].

#### 2.1.1. Synthetic Datasets with Haploid Genome

We utilized the ZS97RS3 rice genome as the haploid reference genome. The reads used were HIFI reads of a homozygous diploid rice, which were synthesized based on the rice genome ZS97 by Hou et al. The coverage of the HIFI reads was 30× Genome assembly was carried out using Hifiasm (v0.16.1) [17], and we followed Hou et al.’s recommendation to employ Yahs (v1.2a.2) [18] for scaffolding the assembly results. This process yielded the required genome sketch. The same reads were used for gap filling.

#### 2.1.2. Real Datasets with Diploid Genome

The dataset consisted of HIFI reads from a segment of the strawberry [19] genome (NCBI project PRJNA801713). The coverage of the HIFI reads was 128×. Genome assembly was performed using Hifiasm, and for chromosome scaffolding in the diploid scaffold, we selected the best-performing software 3D-DNA (v180922) [20], whose efficacy was demonstrated in Hou et al.’s study. These steps led to the creation of the diploid genome sketch. The same reads used for assembly were employed for gap filling. The reference genome for the diploid dataset was sourced from Sun et al.’s publication, which provided the strawberry genome.

#### 2.1.3. Synthetic Datasets with Tetraploid Genome

We used synthesized HIFI reads generated by Hou et al. for a tetraploid genome. The coverage of the HIFI reads was 30×. After assembling using Hifiasm, we employed the software Pin_hic (3.0.0) [21], as described by Hou et al., which exhibited the best performance for tetraploid scaffolds, to scaffold the obtained contigs and generate the tetraploid draft genome. The same reads were used for gap filling. The reference genome was the complete genome of a tetraploid organism used for synthesizing the reads.

### 2.2. Software Running Details

Each software tool is provided with distinct parameters tailored to various data types and ploidy levels. These parameters enable the tools to reduce errors and enhance operational efficiency when filling gaps. Prior to execution, the DENTIST (v4.0.0) software requires configuration through a settings file, encompassing information such as read type, read coverage, and ploidy. LR_Gapcloser, TGS-GapCloser (v1.2.1), and SAMBA necessitate the specification of read type and alignment parameters. On the other hand, PGcloser (v1.2) and FGAP (v1.8.1) involve only fundamental parameters related to alignment and gap length. Among these tools, RFfiller stands out as the simplest, offering only a basic option for the number of threads, with no additional selectable parameters. When running these tools, we strictly adhered to the specified data types and ploidy levels when configuring parameters. In the absence of specific parameters, we followed the default settings provided by the software.

### 2.3. Evaluations

We employed six criteria to assess the quality of the genome assembly after gap filling: NG50, NGA50, misassemblies, genome fraction, completeness, and accuracy. NG50, NGA50, misassemblies, and genome fraction were evaluated using the QUAST (v5.2.0). NG50 is a length, and the collection of all contigs of that length or longer covers at least half the reference genome. NGA50 represents a block length such that all blocks of at least the same length together cover at least 50% of the reference genome. Genome fraction refers to the proportion of the assembled genome compared to the reference genome. Misassemblies refer to the number of positions where the left-side sequence in an assembly differs by more than 1 kbp from the right-side sequence in the reference genome (relocation) or where sequences are positioned on different strands (inversion) or different chromosomes (translocation). We referred to the completeness metrics in Quast-LG [22], proposed our own completeness indicators, and expanded them into accuracy metrics. Both indicators were based on unique k-mer counts, with specific calculation formulas. The formulas for completeness and accuracy are given below:(1)$$v_{completeness}=\frac{S_{refG\_unikmers}\cap S_{filledG\_unikmers}}{S_{refG\_unikmers}}$$
(2)$$v_{accuracy}=\frac{S_{refG\_unikmers}\cap S_{filledG\_unikmers}}{S_{filledG\_unikmers}}$$
where $S_{refG\_unikmers}$ refer to the unique k-mers that reference genome has, and $S_{filledG\_unikmers}$ are unique k-mers in the filled genome. We set all k in k-mer to 21 when we evaluated the completeness and accuracy of the chosen software. This comprehensive approach allowed us to thoroughly assess the gap-filling software performance across different genomic contexts and provide a robust evaluation of genome assembly quality after gap filling.

### 2.4. Runtime and Memory Usage

We configured all software with 32 threads and recorded the runtime and maximum memory usage for each experiment. With the exception of one software that exceeded the memory limit, all gap fillers were run on a server equipped with 2 AMD EPYC 7H12 64-Core CPUs and 1024 GB of RAM. The software that exceeded the memory limit was executed on an Inspur Cluster Engine Linux cluster at the Agricultural Genomics Institute at Shenzhen, Chinese Academy of Agricultural Sciences. This cluster comprises six main nodes, each equipped with 80 CPUs and 3 TB of memory.

## 3. Results

We conducted a comprehensive set of experiments on seven gap fillers, encompassing various ploidies and data generation methods.

### 3.1. Experiments on Synthetic Haploid Datasets

All gap fillers were evaluated using synthetic haploid HIFI reads. The experiments utilized reads identical to those used in the assembly. We performed a comprehensive assessment of these software tools based on the six evaluation metrics mentioned in the Methods section. For the haploid dataset, except for RFfiller and TGS-GapCloser, the memory requirements for each software were quite similar. RFfiller could efficiently fill over 99% of the gaps in the shortest time and with the lowest memory usage. TGS-GapCloser, while using most memory, could also effectively fill over three-quarters of the gaps. A less effective software was DENTIST, which consumed a considerable amount of time and could only fill a minimal number of gaps. The software with the least favorable performance was SAMBA, as it failed to successfully fill any gaps, despite its minimal time and memory consumption (Table 1).

DENTIST was identified as the highest-performing software based on the evaluation metrics of QUAST, as well as on the completeness and accuracy metrics provided. Despite filling a comparatively smaller quantity of gaps, DENTIST guaranteed the precision of the sequences it filled, thereby minimizing potential risks in subsequent genome analysis. Even though other software tools may surpass DENTIST in the number of gaps filled, according to the evaluation from Quast and based on our integrity and accuracy metrics, a portion of the filled gaps by these tools contained errors. For instance, RFfiller, despite addressing the majority of the gaps, resulted in a substantial number of misassemblies, which contributed to a decline in genome accuracy and diminished the overall quality of genome assembly. The reason for SAMBA’s exceptionally high completeness and low accuracy lies in its process of scaffold fragmentation, followed by the reconstruction of contigs using long reads. While this approach increased the correctly assembled portions, it concurrently elevated the incidence of misassemblies, contributing to a reduction in overall accuracy (Figure 1). For synthetic haploid datasets, considering a comprehensive assessment of these metrics, we highly recommend DENTIST. If considering a balance between accuracy and closure rate, FGAP would be preferred. While it may exhibit a lower closure rate, the gaps it filled tended to have relatively higher accuracy, which ultimately enhanced the overall quality of genome assembly. For some software, such as LR_Gapcloser, SAMBA, RFfiller, DENTIST, FGAP, and TGS-GapCloser, when evaluated using QUAST, the values for NG50 and genome fraction were quite similar. This similarity arose because the genome itself, after scaffolding, exhibited high continuity and completeness, and the impact on the overall integrity and continuity of the genome after gap filling was minimal. Therefore, relying on a single metric is insufficient for the effective evaluation of a software tool. It is crucial to consider a comprehensive set of metrics to obtain more accurate comparative results.

### 3.2. Experiments on Real Diploid Datasets

For the diploid dataset, the use of real HIFI reads with high sequencing depth and large data size posed significant memory challenges, particularly for TGS-GapCloser. Therefore, on the diploid dataset, we executed all software on the Inspur Cluster Engine Linux cluster and recorded their runtime and maximum memory consumption. TGS-GapCloser exhibited the highest memory requirements for the diploid dataset gap filling, exceeding 1 TB. It also had the longest runtime and filled the second-highest number of gaps. Conversely, Rffiller remained the software with the least resource demands, using minimal computational resources to fill the highest number of gaps. DENTIST and SAMBA, two software tools, exhibited similar runtime and memory usage; however, SAMBA filled a greater number of gaps (Table 2).

On the diploid dataset, according to the QUAST evaluation, all software exhibited a relatively high number of misassemblies (Figure 2b). This was primarily attributed to the chromosome scaffolding process. Among the seven software tools, SAMBA demonstrated the most robust performance, achieving consistently high scores in accuracy, completeness, and misassemblies compared to the other tools. This indicated a significant improvement in assembly quality after contig reconstruction by SAMBA. Conversely, the least effective software was Rffiller, despite filling over 99% of the gaps. Evaluation across various metrics revealed a substantial decline in assembly quality, suggesting a considerable number of erroneously filled gaps. The remaining software tools, not involving scaffold fragmentation and remounting during the assembly process, exhibited suboptimal performance in terms of poorly mounted results. Additionally, Pgcloser software discarded some scaffolds during runtime, resulting in lower values for NGA50, NG50, and genome fraction.

### 3.3. Experiments on Synthetic Tetraploid Datasets

On the synthetic tetraploid dataset, we tested all software using HIFI reads identical to those used in genome assembly. The software with the highest closure rate was TGS-GapCloser, even though it utilized a substantial amount of memory. Comparing LR_GapCloser and TGS-GapCloser, TGS-GapCloser prioritizes time efficiency, consuming a considerable amount of space, while LR_GapCloser emphasizes space efficiency, leading to an extended runtime for the sake of conserving space. Among the software tools, DENTIST exhibited the poorest performance in terms of gap closure rate. SAMBA and DENTIST failed to close any gaps, which was possibly attributed to the sequencing depth. Rffiller demonstrated the least resource requirements and closed over 26% of the gaps (Table 3).

Upon evaluating the results of all software using QUAST and custom metrics, we identified FGAP as the top-performing software. It performed exceptionally well across all six metrics. Moreover, compared to DENTIST, which did not fill any gaps, it did not introduce misassemblies, maintaining both accuracy and completeness. However, it is essential to note that Pgcloser exhibited starkly different results in terms of QUAST evaluations compared to the custom metrics, and the loss of sequences resulted in lower QUAST indicators in the evaluation. Since the lost sequences might include portions with misassemblies, the reduction in misassemblies was also compromised. Upon closer examination of the Pgcloser results, we identified a significant loss of genomic sequences after generating results. This led to lower values when assessing completeness and accuracy. The reason for the sequence loss was found to be a lack of compatibility with all platforms, resulting in low portability. This caused certain commands to execute erroneously without returning the correct error status, allowing the program to continue running. With Pgcloser excluded from consideration, the software with the least favorable performance was TGS-GapCloser, which exhibited the highest number of misassemblies. This indicated that, during the gap-filling process, TGS-GapCloser introduced a significant number of errors into the genome, thereby diminishing the overall quality. Similar issues were observed for LR_GapCloser and Rffiller. Although both DENTIST and SAMBA ultimately failed to fill any gaps, SAMBA’s scaffold fragmentation and remounting process led to a decrease in both completeness and accuracy. In contrast, DENTIST, without performing any such operations, represents the level of scaffolding after chromosome mounting (Figure 3).

## 4. Discussion

With the advancement of sequencing technologies and the progress of genome assembly software, assembling a complete genome is no longer an insurmountable task. However, inevitable gaps arise during the genome assembly process. Determining the most effective method and software for gap filling requires extensive experimental testing. We evaluated seven genome gap-filling tools on different datasets using comprehensive metrics. According to our experimental results, the least effective software was RFfiller. Despite its ability to fill over 99% of the gaps, the majority of the filled gaps, as assessed by Quast and our proposed metrics, were erroneous. Upon inspection of the ‘filled_gaps_seq.txt’ file generated by RFfiller, a substantial portion of the filled sequences had lengths identical to the initial ‘N’ lengths in the sequences, which were generated by the scaffolding software and did not accurately represent the true gap lengths. We speculate that RFfiller fills sequences based on gap lengths and is heavily influenced ‘N’ length. Thus, using RFfiller requires prior knowledge of the actual gap lengths, which is challenging to determine during the genome assembly process. FGAP, while filling fewer gaps, adopts a conservative strategy, resulting in higher accuracy based on the six metrics we selected. PGcloser exhibited relatively high accuracy across various metrics; however, the software discarded some sequences during the run; we cannot anticipate the impact of losing these sequences on the quality of genome assembly. Based on our results, this will lead to a decrease in genome integrity. TGS-GapCloser consumed the most resources, and although it filled most of the gaps, its accuracy in gap filling was comparatively low based on our evaluation metrics. All software tools were significantly affected by the sequencing depth of the reads, particularly, SAMBA and DENTIST. Our experimental results suggest that these two tools are ineffective in gap filling when the sequencing depth is low. Despite being designed for 10–30× long-read sequencing, SAMBA’s performance may suffer at lower sequencing depths, potentially compromising the quality of the original assembly. Adequate sequencing depth improves SAMBA’s ability to correct errors introduced by scaffolding, enhancing the overall genome quality. The existing software for gap filling using long reads still have considerable room for improvement to achieve a better balance between introducing misassemblies and effectively filling gaps. In the genome assembly process, it is crucial not only to prioritize completeness but also to consider accuracy. Accuracy is particularly important, as, even if completeness is low, it can be enhanced through manual interventions. However, low accuracy makes it challenging to precisely identify problematic assembly regions.

In addition, the existing evaluation metrics proposed by the current software, including those introduced in our study, pose challenges when comprehensively assessing the quality of genome assemblies. These tools provide only a rough assessment of overall genome quality and lack the capability to discern the quality of specific assembly regions. Additionally, if the initial genome assembly quality is poor, it will result in lower values for all evaluation metrics. Evaluating the gap-filling performance of these assembly software tools, also with the use of our assessment metrics, does not accurately reflect the individual gap-filling success for each gap. Indeed, evaluating single-gap filling is a challenging task. When dealing with short gap sequences, the current approach of assessing overall genome assembly quality may not accurately reflect limited improvements in various metrics and may fail to reveal differences before and after gap filling. For instance, some software obtained very similar values for a single metric, making it challenging to discern differences between them. In such cases, a comprehensive consideration of multiple metrics is required to identify the best-performing software. The current methods for evaluating overall genome assembly quality still lack specificity when it comes to assessing the detailed aspects of genome assembly. There is currently no well-established standard in the existing evaluation tools for the detailed assessment of assembly nuances, and we are actively exploring such criteria.

In conclusion, there is significant room for improvement for the existing long-read gap-filling software. Striking a better balance between mitigating misassemblies and effectively filling gaps remains a challenge. Additionally, the evaluation of genome assembly quality, including gap filling, requires more refined and specific metrics to provide accurate insights into the success of gap filling at both the overall and the individual gap levels.

## 5. Conclusions

In this study, we conducted comprehensive benchmark tests on seven genome gap-filling tools. We evaluated their performance across different ploidies and data types, considering NG50, NGA50, misassemblies, genome fraction, completeness, and accuracy. Our results demonstrated significant variations in the performance of these software tools across different ploidy levels and data types. Therefore, users should carefully choose the most appropriate gap-filling tool based on the specific characteristics of their data.

## Figures and Tables

**Figure 1 genes-15-00127-f001:**
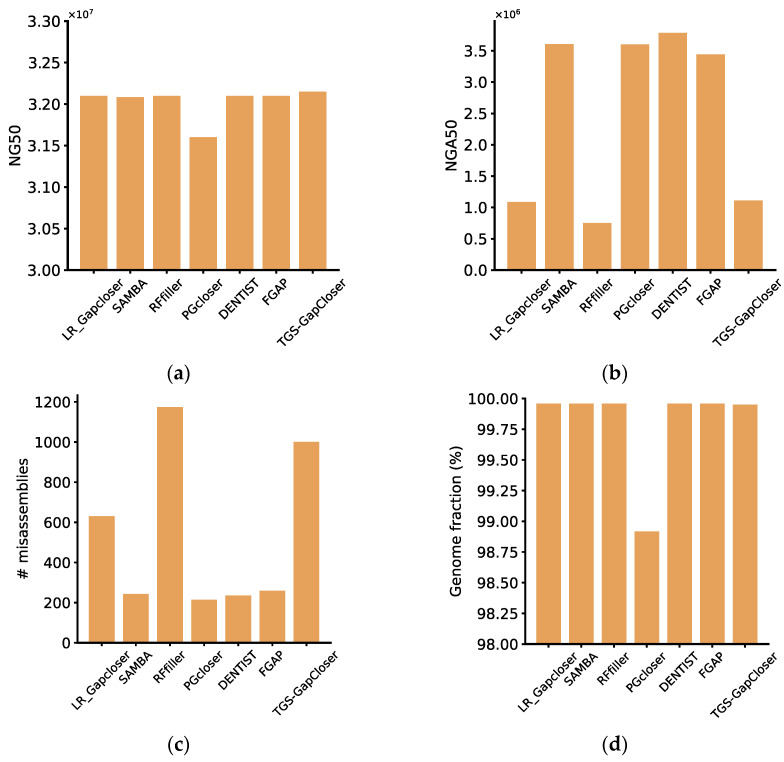
Evaluation of the performance of the gap fillers. (**a**) Scaffold NG50, (**b**) scaffold NGA50, (**c**) misassemblies, (**d**) genome fraction, (**e**) completeness, and (**f**) accuracy for synthetic haploid datasets, calculated by python scripts or reported by QUAST.

**Figure 2 genes-15-00127-f002:**
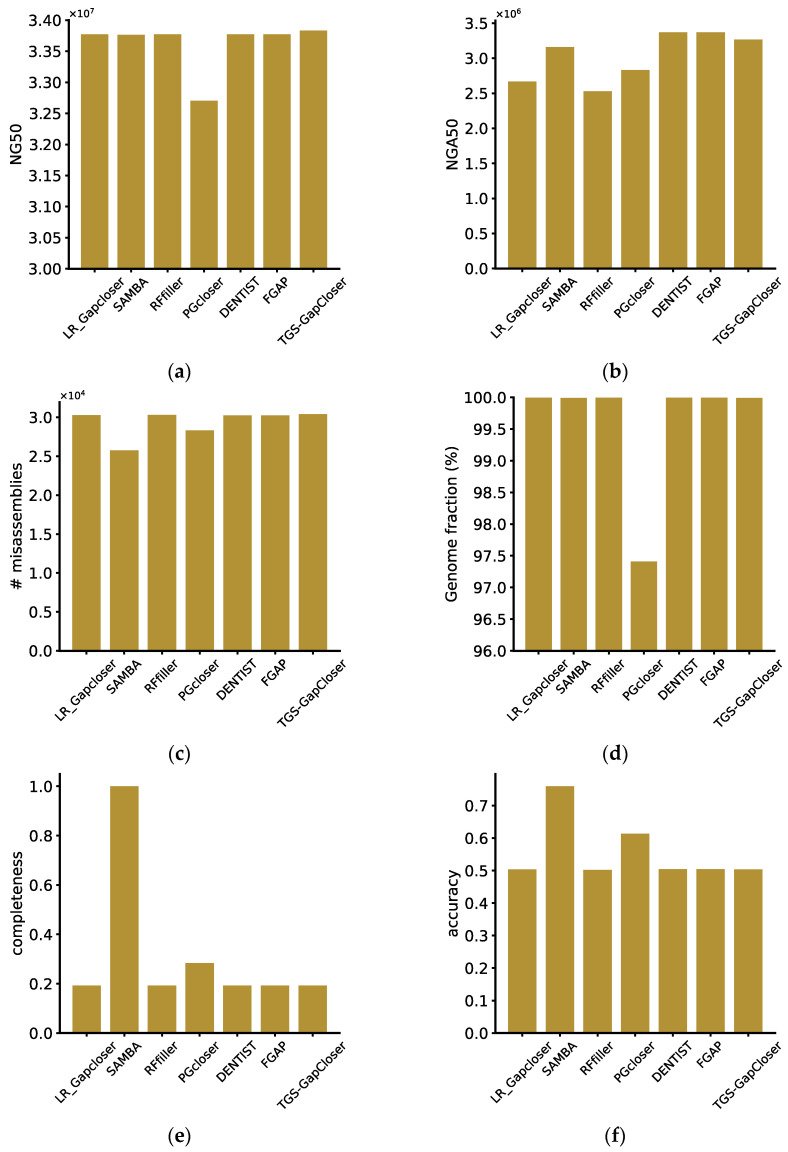
Evaluation of the performance of the gap fillers. (**a**) Scaffold NG50, (**b**) scaffold NGA50, (**c**) misassemblies, (**d**) genome fraction (**e**) completeness, and (**f**) accuracy for real diploid datasets, calculated by python scripts or reported by QUAST.

**Figure 3 genes-15-00127-f003:**
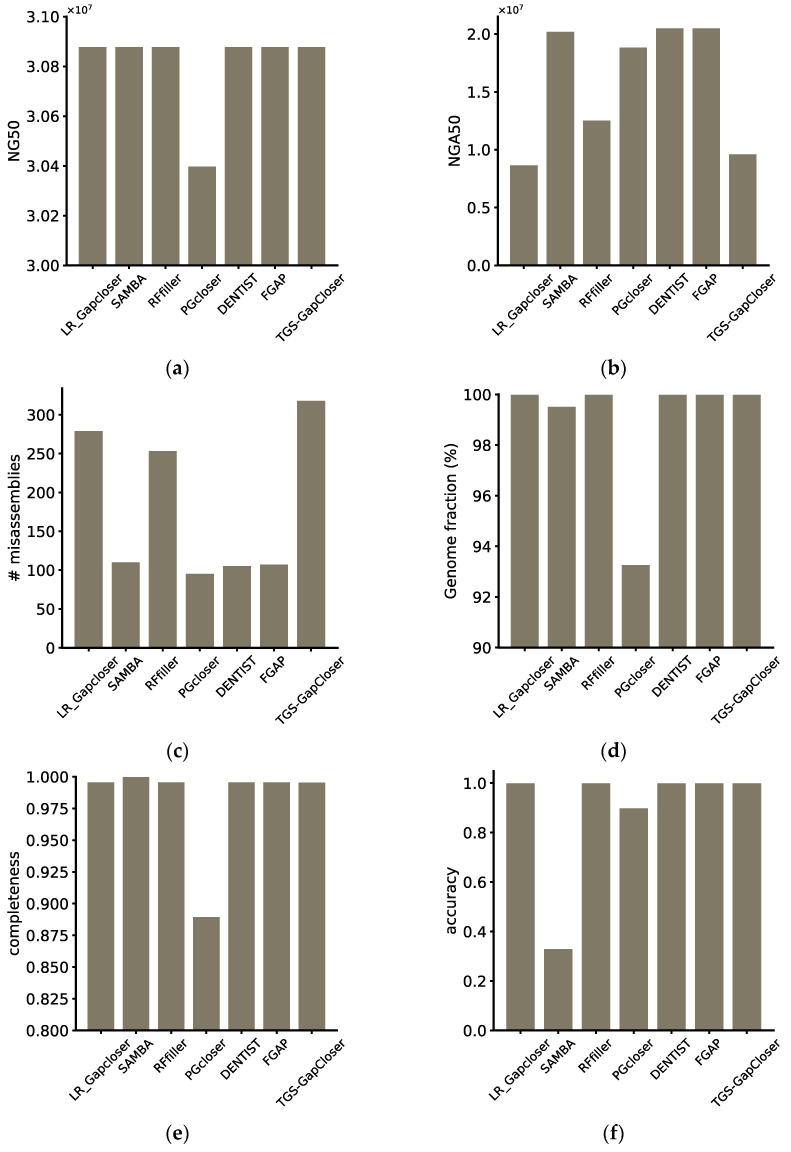
Evaluation of the performance of the gap fillers. (**a**) Scaffold NG50, (**b**) scaffold NGA50, (**c**) misassemblies, (**d**) genome fraction, (**e**) completeness, and (**f**) accuracy for synthetic tetraploid datasets, calculated by python scripts or reported by QUAST.

**Table 1 genes-15-00127-t001:** Number of gaps after gap filling of a haploid dataset, runtime, and maximum memory consumption.

Tool	Number of Closed Gaps	Closed Rate	Runtime (min)	Maximum Memory Consumption (Gb)
LR_Gapcloser	668	49.12%	1243	23.36
TGS-GapCloser	1068	78.53%	510	82.28
DENTIST	5	0.37%	1049	18.16
RFfiller	1359	99.93%	226	2.91
PGcloser	53	3.90%	1104	28.17
FGAP	115	8.46%	126	37.01
SAMBA	0	0	112	6.89

All software were run with 32 threads on the same computer.

**Table 2 genes-15-00127-t002:** Number of gaps after gap filling of a diploid dataset, runtime, and maximum memory consumption.

Tool	Number of Closed Gaps	Closed Rate	Runtime (min)	Maximum Memory Consumption (Gb)
LR_Gapcloser	297	64.71%	9083	64.81
TGS-GapCloser	300	65.36%	14459	1102.01
DENTIST	1	0.22%	3378	17.92
RFfiller	458	99.78%	157	8.03
PGcloser	64	13.94%	6346	84.92
FGAP	13	2.83%	270	74.03
SAMBA	5	1.09%	2199	17.85

All software were run with 32 threads on the same computer.

**Table 3 genes-15-00127-t003:** Number of gaps after gap filling of a tetraploid dataset, runtime, and maximum memory consumption.

Tool	Number of Closed Gaps	Closed Rate	Runtime (min)	Maximum Memory Consumption (Gb)
LR_Gapcloser	224	32.37%	8300	88.36
TGS-GapCloser	242	34.97%	4761	305.38
DENTIST	0	0.00%	6299	24.04
RFfiller	183	26.45%	365	10.74
PGcloser	55	7.95%	7330	117.73
FGAP	35	5.06%	712	94.99
SAMBA	0	0	578	15.28

All datasets were run with 32 threads on the same computer.

## Data Availability

The script that implements the two quality criteria can be obtained at https://github.com/xianjia10/gap-filler_evaluation.git, accessed on 26 December 2023.

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
