# Peer review of "Comprehensive Evaluation of Genome Gap-Filling Tools Utilizing Long Reads"

_genes, 2024, doi:10.3390/genes15010127_

Round 1
Reviewer 1 Report
Comments and Suggestions for Authors
The brief report article “Comprehensive evaluation of genome gap-filling tools utilizing long reads” aims to provide reader a comprehensive evaluation of genome gap-filling tools. The authors evaluated 5 different gap-filling tools using 4 metrics, namely (a) NGA50, (b) misassemblies, (c) completeness and (d) accuracy. For completeness and accuracy, the authors use k-mer count approach for intersections between the reference and gap-filled datasets.
Major Revisions
Some clarifications on definitions of completeness and accuracy needed to correctly interpret the results presented in the paper. Eg. (1-p1) and (1-p2) in formulae would represent the un-intersected data within the denominator sets and need more clarification on definitions or the formulae need updating. Additionally k in k-mers is not mentioned in the submitted version, which may help reader understand the manuscript better.
Comments on the Quality of English LanguageMinor editing is needed
Reviewer 2 Report
Comments and Suggestions for Authors
Dear authors,
I have reviewed your manuscript, which explores the benchmarking of six genome gap-filling tools using long sequencing reads. The paper compares the performance of these tools across various ploidy levels and data generation methods, utilizing QUAST metrics alongside in-house metrics such as completeness and accuracy. The primary conclusion drawn is the existing room for improvement in the field, emphasizing the importance of selecting the most suitable tool based on specific data features. I would like to offer some feedback for improvement:
Minors:
- Consistency in Terminology:
- Ensure consistent usage of terms and vocabularies throughout the manuscript, such as RFfiller: Rffiller, Rffiller, Rfiller, Rfilller (line 46, 61, 137, 138, 149, 180, 213, etc.). Other examples include DENTIST (Dentist) and HIFI (HIFi, HiFi).
- Grammar and Typos:
- Rectify missing spaces (line 76, 92, 159, etc.).
- Address inconsistent font or style (line 80, 87, 88, 92, etc.).
- Correct typos (line 157, 159, etc.).
- Refine sentences requiring improvement (line 68-69, 107, etc.).
Majors:
- Correct Reference:
- Ensure accurate references, particularly for FGAP, with the correct paper being https://pubmed.ncbi.nlm.nih.gov/24938749/.
- Comprehensive Tool Summary:
- Expand the manuscript by including other gap-filling tools like Sealer, Gap2Seq, GAPPadder, SAMBA, providing a more comprehensive table outlining current tools and the inclusion criteria for this study.
- Benchmarking of In-House Metrics:
- Validate the in-house metrics (completeness and accuracy) independently and discuss the impact of K-mer selection on their performance.
- Software Parameters:
- Clarify how software parameters were determined during benchmarking, considering that default settings may not be optimal for every ploidy level and data generation method. Address whether RFfiller has parameters to mitigate the over gap-filling problem.
- Effectiveness of Proposed Metrics:
- Address the concerns regarding the effectiveness of the newly proposed metrics listed below.
- According to Table1, if DENTIST only fills 5 gaps, how can it be ranked as the highest performing tool in terms of the completeness metric?
- The completeness value barely changes across different tools according to Figure 1C. And it is hard to believe that, based on Figure 2C and 3C, there are 5 tools that outcome nearly identical completeness values.
- For completeness and accuracy, there is a 10-fold difference according to Figure 1C and Figure 2C, what does that suggest?
- Tool Recommendations Based on Accuracy:
- Even assuming that the newly proposed metrics are legit, I have to challenge the fairness of tool recommendations based on accuracy metrics (line 151-153), given that all tools exhibit very high accuracy values (> 60), which translate to nearly identical p2 values for all tools.
- Misassembly Metric and NGA50 Values:
- Given that DENTIST filled 0 gap according to Table 3, which can be served as control to benchmark all metrics, please explain the even lower misassembly metric value for PGcloser compared to DENTIST in Figure 2b and the meaning of lower-than-DENTIST NGA50 values as observed in Figure 2a.
Thank you for your attention to these suggestions.
Comments on the Quality of English LanguageOverall, the English in the submitted manuscript is clear and communicates the intended message effectively.
Round 2
Reviewer 2 Report
Comments and Suggestions for Authors
Thank you for submitting your revised manuscript and cover letter, addressing my initial comments. However, there are a few areas that require further attention:
- Typos and formatting:
- Line 81: There is an extra space before comma.
- Line 133: A space is missing after the colon.
- Line 124, 310, 312, 314: "SMABA" should be "SAMBA".
- Lines 206-207: Redundant sentences.
- Lines 206-207: Redundant sentences.
- Line 315: A space is missing after the period.
- Lines 322-323: Further editing is required.
- "Genome fraction" and "Genome Fraction" are used inconsistently.
- Please provide justification for selecting 21 as the k-mer size (line 148).
- The behavior of the "completeness" metric appears controversial, as SAMBA closes zero gaps according to Table 1, yet it achieves the highest "completeness" in Figure 1e. Also, a "completeness" value of 1 may be misleading, as it could suggest that gap-filling is complete, whereas in the case of SAMBA, it may indicate that zero gaps have been filled. Consider modifying the terminology to accurately convey the extent of gap-filling achieved. Similarly, for the "accuracy" metric, reconsider its terminology for clarity.
Please fix the typos and other issues mentioned above.
